# Relating Geotechnical Sediment Properties and Low Frequency CHIRP Sonar Measurements

Reem Jaber [1,*,†], Nina Stark [1,2], Rodrigo Sarlo [1], Jesse E. McNinch [3] and Grace Massey [4]

1 Jr. Department of Civil and Environmental Engineering, Virginia Tech, The Charles Edward Via, 200 Patton Hall, Blacksburg, VA 24061, USA; nina.stark@essie.ufl.edu (N.S.); sarlo@vt.edu (R.S.)
2 Engineering School of Sustainable Infrastructure and Environment (ESSIE), University of Florida, 265H Weil Hall, Gainesville, FL32611, USA
3 US Army Corps of Engineers, Detroit District, Detroit, MI 48226, USA; jesse.e.mcninch@usace.army.mil
4 Virginia Institute of Marine Science, Gloucester Point, VA 23062, USA; grace.massey@vims.edu
* Correspondence: reemj@vt.edu
† Current address: Geosyntec Consultants, Acton, MA 01720, USA.

**Abstract:** Low frequency acoustic methods are a common tool for seabed stratigraphy mapping. Due to the efficiency in seabed mapping compared to geotechnical methods, estimating geotechnical sediment properties from acoustic surveying is attractive for many applications. In this study, co-located geotechnical and geoacoustic measurements of different seabed sediment types in shallow water environments (<5 m of water depth) are analyzed. Acoustic impedance estimated from sediment properties based on laboratory testing of physical samples is compared to acoustic impedance deduced from CHIRP sonar measurements using an inversion approach. Portable free fall penetrometer measurements provided in situ sediment strength. The results show that acoustic impedance values deduced from acoustic data through inversion fall within a range of ±25% of acoustic impedance estimated from porosity and bulk density. The acoustic measurements reflect variations in shallow sediment properties such as porosity and bulk density (~10 cm below seabed surface), even for very soft sediments ($s_u < 3$ kPa) and loose sands (~20% relative density). This is a step towards validating the ability of acoustic methods to capture geotechnical properties in the topmost seabed layers.

**Keywords:** CHIRP; seabed layer; shallow water; inversion

## 1. Introduction

Offshore engineering activities, including coastal infrastructure development, renewable energy, and naval applications, in addition to coastal hazards studies require a detailed site investigation and knowledge of seabed stratigraphy and the sediment properties of the identified strata. Geotechnical sediment properties are often obtained using Cone Penetration Testing (CPT) which allows to estimate strength parameters, or relative density, which are then typically confirmed by physical sediment samples and sediment cores [1,2]. However, CPT can be cost and time-intensive and might not be able to capture the spatial variability in soils with single point measurements or seabed surface properties [1]. Thus, geophysical methods, and specifically, low frequency acoustic profiling is a common tool for mapping seabed strata [3].

Geoacoustic surveying is a key component of offshore site investigation for a variety of offshore engineering, resource exploration, research, and naval applications. Different techniques and systems exist for different applications and data product targets even within the topic of seabed classification and characterization [4,5]. This study is focused on investigating seabed sediment properties at a depth of one meter or more below the seabed surface. Low frequency geoacoustic surveying including parametric sub-bottom profilers and CHIRP (Compressed High Intensity Radar Pulse) sonar are most commonly suggested for such tasks, e.g., [6–8]. This study employed CHIRP sonar. Since sound wave

propagation within the seafloor is dependent on the textural properties of sub-surface sediments, continuous efforts have investigated estimating sediment properties such as sediment type, grain diameter, porosity, density, roughness, and others, from low-frequency acoustic surveying [9,10]. However, due to the complicated multi-variable nature of the problem and the challenges associated with acoustic sensitivity to system settings and high spatial variability of soil, developments of direct correlations between acoustic measurements and sediment properties still face limitations when broadly applied to field conditions [3,11,12]. Different approaches were proposed to quantitatively derive geotechnical properties from acoustic parameters using CHIRP sonars. In recent decades, seismic inversion was developed for subsurface acoustic models. It includes a forward model capable of predicting seismic data [13,14]. It was initially applied to hydrocarbon reservoir characterization and was later used for high resolution/shallow water reflection data based either on waveform inversion (pre-stack or post-stack impedance inversion) or based on intrinsic attenuation that often requires the calculation of seismic quality factors [14]. These procedures promise a rapid and efficient mapping of sediment type and properties even for shallow layers (up to 1 m below the seabed), minimizing the need for extensive sample collection and the challenges associated with it [7,11]. However, there is still a need to assess the performance of acoustic inversion for deriving sediment properties for different soil types and conditions, specifically in shallow water environments (<10 m of water depth), and for shallow seabed strata relevant for coastal sediment dynamics [5,11]. Furthermore, there is still work required to better understand and link acoustic measurements with strength-related sediment properties which can be measured in situ [10], and to combine acoustic measurements with geotechnical measurements such as Cone Penetration Testing (CPT), or free fall penetrometers [1,11,15].

This study investigated the use of CHIRP sonar for shallow (sediment depth < 1 m) seabed characterization in shallow-water coastal environments in conjunction with geotechnical in situ testing. The paper applies an acoustic impedance inversion approach to three sediment types, combined with portable free fall penetrometer (PFFP) measurements and sediment laboratory characterization. The goal is to investigate the relationships between acoustic impedance and geotechnical properties measured by two instruments suitable for deployment in areas of active sediment dynamics. Also, to the authors' best knowledge, the inversion approach used has not been applied to seabed surface sediments (<1 m below seafloor) (mbsf) and in shallow waters (on the order of meters).

## 2. Methodology

The results presented in this paper are based on datasets collected from the York River estuary, Virginia, USA during two field surveys conducted in May and June 2021, featuring sites with different soil types and shallow water depths ranging from 2 to 4.5 m: Site 1 ($37.3398°$, $-76.6314°$) is predominantly fine-grained with a water depth of ~2.3 m; Site 2 ($37.3436°$, $-76.6227°$) is predominantly mixed sediments with ~58% fines and a water depth of ~4.5 m; and Site 3 ($37.2463°$, $-76.444°$) is predominantly coarse-grained (with a median grain size of 0.4 mm) and has a water depth of ~3.3 m (Figure 1). The instruments and processing techniques used are described in this section.

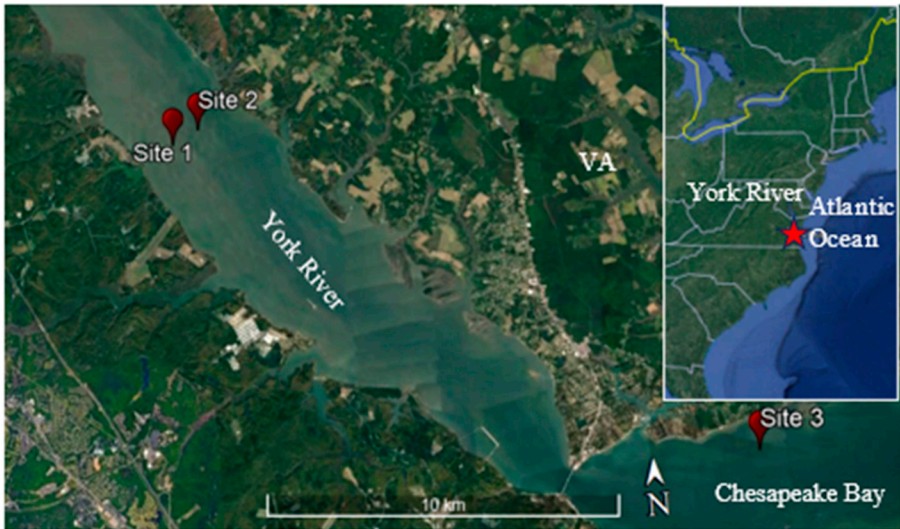

**Figure 1.** Locations of Sites 1,2, and 3 (Map data: Google, SIO, NOAA, US Navy, NGA, GEBCO).

*2.1. CHIRP Sonar and Seismic Inversion*

A CHIRP sonar (SyQwest Stratabox) mounted on the side of a boat was used to collect low frequency acoustic measurements using a firmware data acquisition software (Stratabox HD manual 2016). It produces an 8–12 kHz source sweep and was operated in a low energy mode (300 Watts) to display upper surface sediments ($\leq$5 m below the estuary bed). The time and strength of the return signal were then processed and presented in terms of the reflected signal at different depths below seabed. Stationary CHIRP measurements were recorded at each of the sites, representing a local average. Post-processing of the CHIRP data was based on the inversion approach described in following subsection.

Seismic inversion is a process used to obtain a quantitative subsurface sediment profile from high resolution seismic data. It is solved by reducing the error between recorded and synthetic CHIRP sonar traces dependent on sediment properties and acoustic signal characteristics, through different optimization techniques [3,10,13,14]. The inversion approach can represent a pre-stack (full waveform inversion based on the P and S-wave velocity model and attenuation model) or a post-stack model (impedance inversion based on an impedance model). Post-stacking includes different geostatistical methods to estimate sediment properties including model-based inversion, colored inversion, band-limited impedance [12]. This paper applies a post-stack model-based inversion approach following the procedure suggested by [10]. The approach is summarized in Figure 2 and is based on minimizing the error between synthetic and recorded CHIRP amplitude envelopes. The synthetic data is obtained through convolution of the reflectivity calculated from the initial acoustic impedance step function (Equation (1)) with the source waveform which depends on the CHIRP source sweep (in this paper: 8–12 kHz sweep).

$$R = \frac{Z_2 - Z_1}{Z_2 + Z_1} \tag{1}$$

where $R$ is the reflection coefficient, $Z_1$ and $Z_2$ are the acoustic impedances of the two media. The synthetic CHIRP trace is then cross correlated with the source sweep to generate the synthetic CHIRP signal, which is compressed using a Hilbert transform and the amplitude envelope is extracted. It should be noted that 1-D inversion is valid in this case, because the upper surface sediment structure is considered layered, and the CHIRP sonar is sensitive mainly to sediment density and impedance [10]. The inversion and optimization were performed using an in-house developed code written in Matlab that is available at [16]. The inversion is applied to a part of the trace (upper ~1 m of sub-surface sediments) where box core samples or disturbed grab samples were extracted and utilized to compute acoustic impedance values. The measured acoustic impedances from the physical samples were

compared to the acoustic impedance backcalculated from the fitted CHIRP envelope to verify the ability of the inversion process to predict the measured impedance. The average sound velocity used initially for time-depth conversion of the acoustic measurements is 1510 m/s based on conductivity, temperature, and depth (CTD) measurements during the survey. However, the acoustic impedance profile derived from the inversion process is representative of changes in sediment density and associated sound velocity.

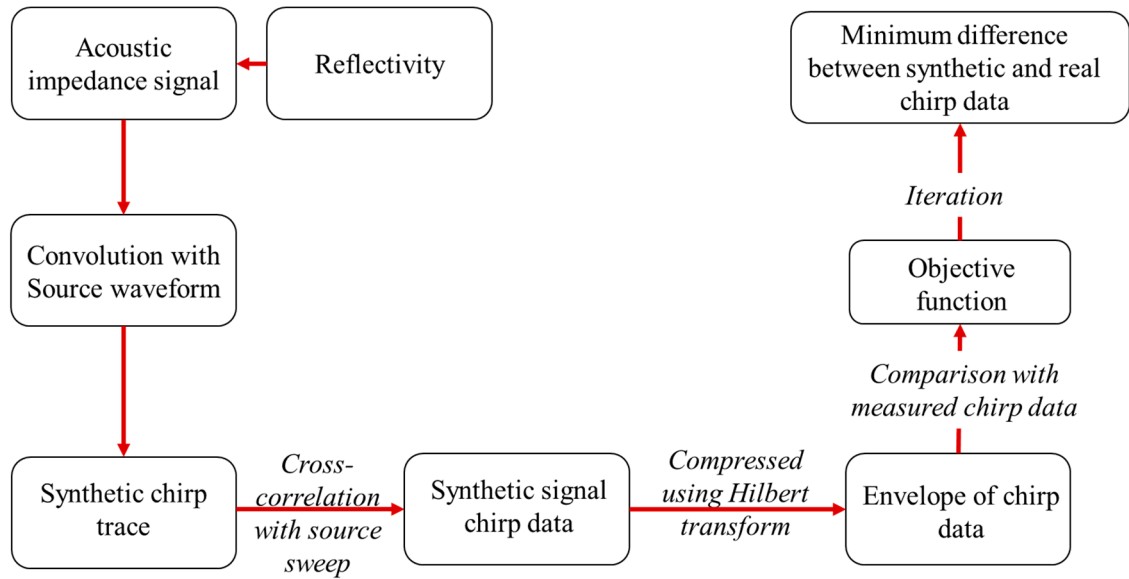

**Figure 2.** Flowchart summarizing the steps of inversion approach adopted.

*2.2. Physical Samples*

　　Box core sediment samples were collected for sites S1 and S2 revealing fine-grained sediments, and disturbed grab samples were collected using a small-size grab sampler at S3 revealing coarse-grained sediments. Box core samples were tested in the laboratory to determine grain sizes, water content, bulk density according to the American Society for Testing and Materials standards ASTM D1140, D6913, D2216, and D7263 [17–20], respectively. Porosity was also measured directly when box core samples were collected or deduced from phase relationship using water content measurements. Box core samples were analyzed in vertical segments of ~10 cm each. Sediment properties such as the bulk density $\rho_b$ and the porosity $n$ were then measured for each depth range. To validate direct measurements of porosity values, values were compared with those estimated from water content. For example, porosity values reported at S1 for layer 1 (8–19 cm), were 0.78 and 0.73 based on water content derivations and direct measurements, respectively, and 0.76 and 0.78 for layer 2 (19–29 cm). Since the error is <7% between these two methods of measurements, porosity was measured directly when possible (box core samples were extracted) and using water content otherwise (when disturbed samples were collected). Disturbed samples were tested for grain size and water content only.

　　Acoustic impedance $Z_i$ was estimated from sediment samples following the LeBlanc et al. (1992) [9] approach, using compressional wave velocity (dependent on porosity) and bulk density (Equation (2)).

$$Z = \rho_b c_b \tag{2}$$

where $\rho_b$ is the bulk density of sediment and $c_b$ is compressional wave velocity. A direct measurement of sound velocity would have been preferable but is also difficult in itself. Therefore, $c_b$ was estimated based on relationships to textural sediment properties and compressibility parameters as suggested by [9].

Acoustic impedance was also calculated using [11] based on porosity, *n*, only, as shown in Equation (3). This equation was based on sandy and clayey soil for shallow and deep-water areas.

$$Z_s = 1.315 \times 10^{-4} \times n^4 - 3.776 \times 10^{-2} \times n^3 + 4.201 \times n^2 - 2.450 \times 10^2 \times n + 8.603 \times 10^3 \qquad (3)$$

Portable Free Fall Penetrometer (PFFP)

The PFFP *BlueDrop* is an in situ geotechnical site characterization tool for preliminary site characterization and rapid sediment mapping [21,22]. It is equipped with vertical accelerometers measuring between ±1.7 and ±250 g (with *g* being the gravitational acceleration) which continuously measure deceleration as the penetrometer penetrates the seabed. The penetrometer is described in more detail in [8,23]. There have been continuous efforts to estimate geotechnical sediment properties for different soil types from the PFFP, the latest being the work presented in [16], proposing a generalized framework to estimate geotechnical sediment properties such as relative density and friction angle for coarse-grained sediments (with % fines < 20) and undrained shear strength for fine-grained sediments (with % fines > 20) from the PFFP measurements. Following the framework mentioned above, the undrained shear strength ($s_u$) for sites S1 and S2, and the relative density for site S3 as described in [16,24,25] was estimated from PFFP measurements. For estimating $s_u$, a quasi-static bearing capacity (*qsbc*) is first calculated from the deceleration values measured by the PFFP using either a logarithmic or power law strain rate correction and then divided by a cone factor to calculate the undrained shear strength. A logarithmic strain rate correction factor *k* of 0.13 was used to calculate the *qsbc,* within the range of 0.1–0.15 recommended by [16] and a cone factor $N_{kt}$ of 10 was used to calculate $s_u$, as recommended in Mayne and Peuchen (2018) [26] and further supported by [25].

### 3. Results

The results obtained from laboratory testing, geotechnical testing using the PFFP, and acoustic impedance deduced from CHIRP data using the inversion approach, are presented in this section.

### *3.1. Laboratory Testing*

Site 1 (S1) is composed of fine-grained sediments with 98% fines and a plasticity index *PI* of 14 and 22 for segments 1 and 2 (each ~10 cm long), respectively; S2 was characterized by 58% fines and a *PI* of 20; and S3 is predominantly coarse-grained, classified as poorly graded sand with less than 1–3% fines. In the case of S3, where only disturbed samples could be collected, porosity was estimated based on the water content measured using phase diagram relationships. The first 6–7 cm for S1 and S2 were lost during collection and transport.

The wet bulk density and porosity results were then utilized to calculate the acoustic impedance shown in Table 1 following LeBlanc et al.'s (1992) [9] approach, which will be referred to as $Z_s$ from hereon. The acoustic impedance results were ~2.24 × 10⁶ kg/m²s–2.33 × 10⁶ kg/m²s for S1 and 2.19 × 10⁶ kg/m²s for S2. For S3, the acoustic impedance was also calculated using the relationship shown in Equation (2). The synthetic acoustic impedance Zs for S3 was calculated based on the equations presented in [9,11], respectively. The results yielded an agreeable match with values of 2.78 × 10⁶ kg/m²s and 2.93 × 10⁶ kg/m²s, respectively, representing less than 6% deviation.

**Table 1.** Sediment properties obtained from sediment samples collected.

| Site | Soil Type | Depth Range (cm) | Fines (%) | Water Content $w$ (%) | Bulk Density $\rho_b$ (kg/m$^3$) | Porosity $n$ (Unitless) | Acoustic Impedance $Z_s$ (kg/m$^2$s) | LL (%) | PI (%) |
|------|-----------|------------------|-----------|-----------------------|----------------------------------|-------------------------|--------------------------------------|--------|--------|
| S1 | Fine-grained soil | 8–19 | 98 | 130 | 1710 | 0.73 | $2.33 \times 10^6$ | 61 | 47 |
| | | 19–29 | 98 | 119 | 1670 | 0.78 | $2.24 \times 10^6$ | 52 | 31 |
| S2 | Mixed soil | 7–20 | 56 | 96 | 1535 | 0.74 | $2.19 \times 10^6$ | 42 | 21 |
| S3 | Coarse-grained soil | 7–22 | 0.8 | 39 | 1814 | 0.51 * | $2.78 \times 10^6$ | - | - |

* Porosity for Site 3 (predominantly sandy site) was derived from water content measurement.

### 3.2. Portable Free Fall Penetrometer

Figure 3 shows the average deceleration profiles based on five PFFP deployments at each of the sites. The average maximum deceleration measured by the PFFP at S1 is 3.5 g (with g being gravitational acceleration), 5.6 g at S2, and 8.6 g at S3, recorded at sediment depths of ~75, 45, and 14 cm, respectively. Larger deceleration values reflect higher resistance applied against the PFFP by the soil, and therefore, larger deceleration values are often associated with stiffer soils, higher sand %, and a smaller penetration depth. This agrees well with the maximum deceleration values observed at the three sites, where S1 (the site with finest particles and lowest sand %) recorded the lowest deceleration value at the highest penetration depth (Figure 3, red line) and S3 (the site with the coarsest particles and highest sand %) recorded the highest deceleration value at the shallowest depth (Figure 3, black line).

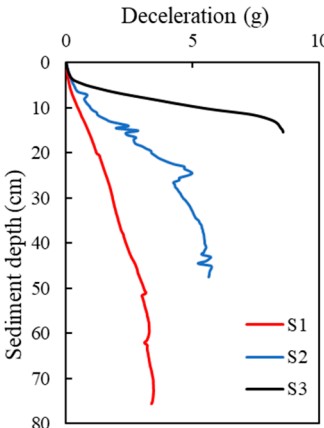

**Figure 3.** Deceleration profiles measured by the PFFP for the three sites.

Based on the framework presented in [16], and as confirmed by the % fines results, S1 is considered a fine-grained soil (penetration depth > 20 cm and % fines > 20), S2 has less % fines than S1 and less % sand than S3 (penetration depth > 30 cm and % fines > 20), and S3 is considered a coarse-grained soil (penetration depth < 20 cm and % fines < 20). Undrained shear strength profiles $s_u$ were determined for S1 and S2 (Figure 4) and a relative density profile $R_d$ was determined for site S3 (Figure 4) calculated from the deceleration profiles in Figure 3. The solid-colored lines (red for S1 and blue for S2) in Figure 4 reflect $s_u$ profiles with a logarithmic strain rate correction ($k = 0.13$). S1 recorded a maximum undrained shear strength of ~3 kPa at a depth of ~75 cm (red solid line in Figure 4), whereas S2 recorded a higher maximum of ~5 kPa at a depth of ~45 cm (blue solid line in Figure 4). It should be noted that the different strain rate corrections methods introduce a maximum deviation of ~3 kPa in $s_u$ values. Estimated relative density values for S3 shown in Figure 5 were

on average ~20% at a maximum sediment depth of ~20 cm. As this approach requires an assumption of the coefficient of consolidation, error bars of ±10% were added to include the range of $R_d$ values estimated for different coefficient of consolidation values.

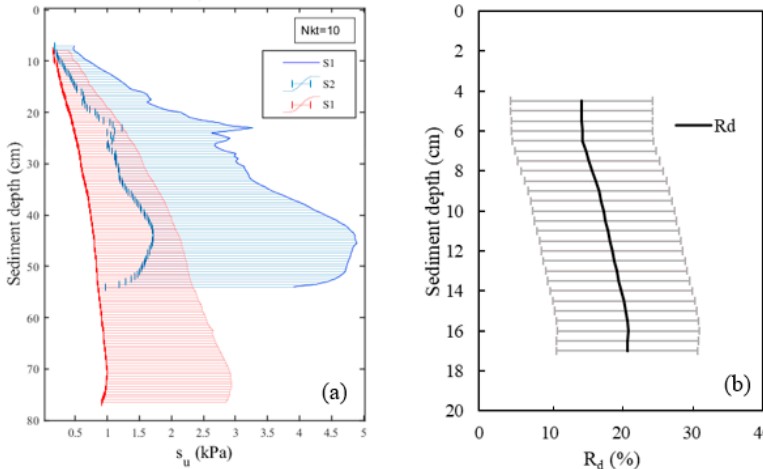

**Figure 4.** (**a**) Undrained shear strength profiles from PFFP measurements for S1 and S2 based on *qsbc* using logarithmic strain rate correction (k = 0.13) and (**b**) relative density ($R_d$) values estimated for S3 along the sediment depth using PFFP measurements (black solid line) with errors bars of ±10%.

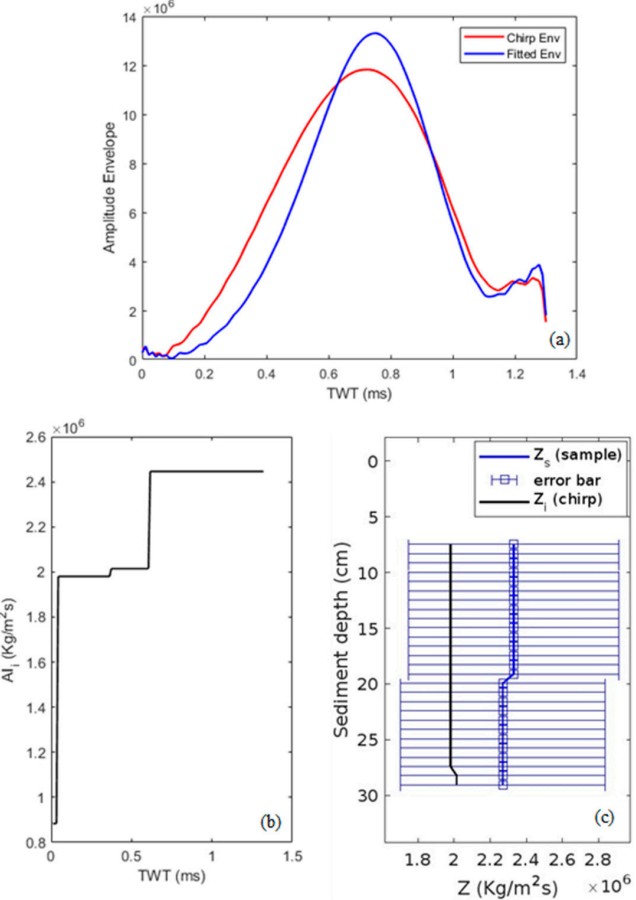

**Figure 5.** (**a**) Measured CHIRP sonar envelope (red) vs. fitted synthetic envelope (blue) for S1. (**b**) Final acoustic impedance model used to produce the fitted envelope as a function of two-way time and (**c**) deduced acoustic impedance from CHIRP $Z_i$ (black line) and acoustic impedance measured from sediment properties $Z_s$ (blue line with errors bars) with sediment depth.

### 3.3. CHIRP Sonar & Seismic Inversion

The inversion approach is applied to the stationary data recorded by the CHIRP sonar as described in Methodology Section. The synthetic CHIRP trace envelope is produced following the inversion approach, which is based on the acoustic impedance model hereby referred to as $Z$. The iterative process was ended when a minimum value of the objective function (difference between synthetic and recorded CHIRP envelope) was achieved given the problem constraints and the level of accuracy provided. For this paper, the inversion is performed on the upper 1 m of the seabed penetration in the CHIRP reflection trace but truncated to focus on the upper 20–30 cm only as the sediment properties were measured for the upper 20–30 cm range. Figure 5a shows the envelope extracted from the recorded CHIRP trace (red line) for S1 compared to the best-fit envelope extracted from synthetic data (blue line) based on the acoustic impedance model $Z_i$ shown in Figure 5c. The variation in $AI_i$ as a function of two-way time (TWT) of the signal ranges between $1 \times 10^6$ and $5 \times 10^6$ kg/m$^2$s in a span of ~1.3 ms of TWT or within the uppermost meter of the seabed. However, since $Z_s$ is estimated from sediment properties in the upper ~30 cm regarding the processing of CHIRP in Table 1, the variation of $Z_i$ (black line) in the top ~30 cm is emphasized in Figure 5c to match the length of $Z_s$ (blue line) as mentioned earlier. To account for the uncertainties in sample collection, testing, and the method used to estimate $Z_s$, an error bar of regarding the processing of CHIRP $\pm 25\%$ is added, described in more detail in the next section (Figure 5c).

The deduced acoustic impedance $Z_i$ from the inversion approach suggests slightly lower values than $Z_s$. Nevertheless, it falls well within the range of the uncertainty of the acoustic impedance based on sediment properties (Figure 5c). The CHIRP vs. synthetic envelope and the resulting $Z_i$ profiles for S2 and S3 are shown in Figures 6 and 7, respectively. A close match between $Z_i$ and $Z_s$ was achieved for S2 with slightly lower values for $Z_i$ (Figure 6). And similarly, a good match was achieved at S3, with $Z_i$ being slightly larger than $Z_s$ (Figure 7). In summary, $Z_i$ yielded slightly lower values of $Z_s$ for fine-grained sites with more mismatch with the presence of a higher fines content, while $Z_i$ yielded slightly higher values than $Z_s$ for the coarse-grained site. In all cases, $Z_i$ and $Z_s$ matched well within the range of uncertainty from sediment sampling.

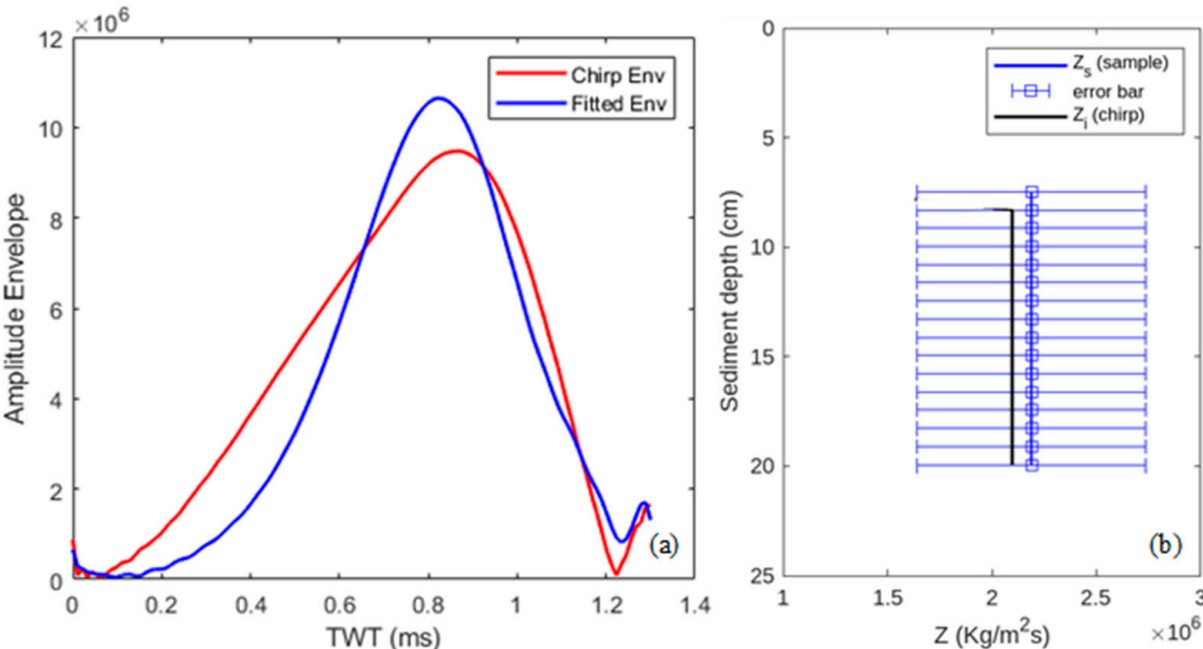

**Figure 6.** (**a**) CHIRP envelope vs. fitted envelope for S2 and (**b**) estimates of $Z_s$ (blue line) and $Z_i$ (black line) with sediment depth.

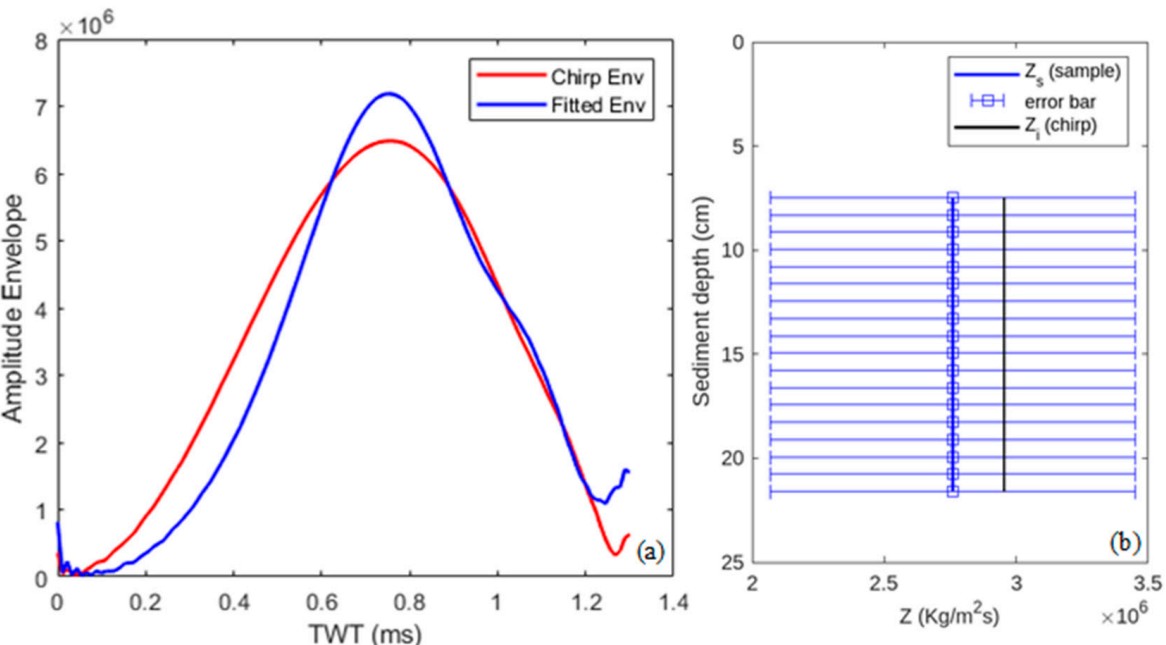

**Figure 7.** (**a**) CHIRP envelope vs. fitted envelope for S3 and (**b**) estimates of $Z_s$ (blue line) and $Z_i$ (black line) with sediment depth.

Figure 8 compares the acoustic impedance values deduced from the CHIRP sonar $Z_i$ (triangles) and estimated from the sediment properties $Z_s$ (circles) with the (a) average maximum deceleration values measured by the PFFP at the respective sites and the $s_u$ values deduced from PFFP measurements. For sites with fine-grained sediments (S1 and S2), a linear trend is apparent based on sediment types as shown by the purple dashed line (with $R^2$ of 0.74); where the deceleration increased with acoustic impedance values (shifting from red color to black color). However, the acoustic impedance deduced from the CHIRP sonar and estimated from sediment properties appeared not related to changes of $s_u$ within range of 0.3–1.7 kPa (Figure 8b) which all can be considered very soft sediments.

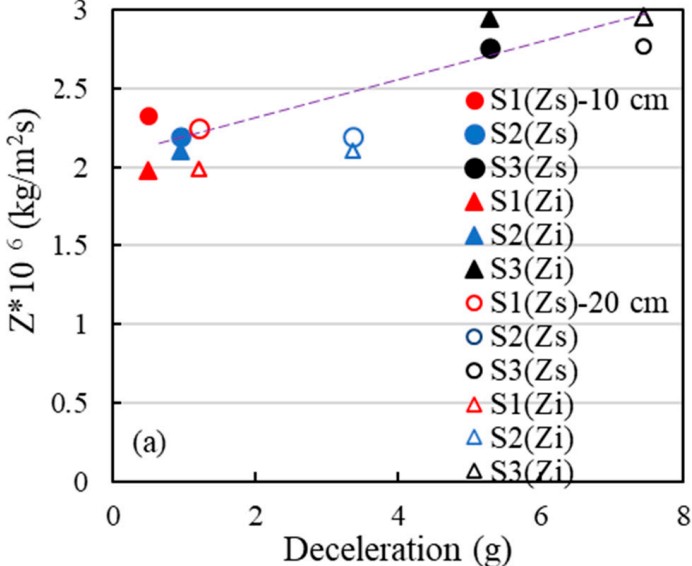

**Figure 8.** *Cont.*

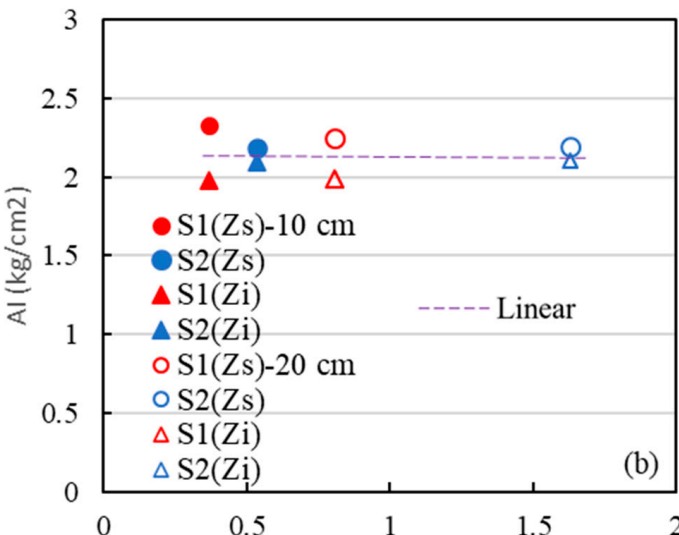

**Figure 8.** Acoustic impedance from sediment samples (circle shaped) and CHIRP measurement (triangle shaped) vs. (**a**) deceleration from PFFP and (**b**) $s_u$ from PFFP at depths of 10 cm (filled) and 20 cm (unfilled) for S1 (red color), S2 (blue color) and S3 (black color). The purple dashed line is a trendline.

## 4. Discussion

The CHIRP sonar used in this study sweeps from 8 to 12 kHz. The acoustic data was processed using a post stack model-based inversion approach to find the best-fit acoustic impedance model that would produce a fitted amplitude envelope that matches the measured CHIRP amplitude envelope best following the approach suggested by [10]. The deduced acoustic impedance $AI_i$ for the three sites fell well within the range of $Z_s$ estimated from sediment properties and associated uncertainties (Figures 5–7). Uncertainties affecting $Z_s$ considered were: (1) sample disturbance from collection and transportation, (2) uncertainties from measuring porosity, bulk density, and water content, this includes possible loss of moisture during sample storage, and (3) uncertainties within the calculation method of the acoustic impedance. Ref. [27] reported errors up 10% in friction angles measurements, and up to 14% in shear strength due to sampling only. This paper adopts LeBlanc et al. (1992) [9] approach to calculate $Z_s$ which depends on $\rho_b$ and $n$, however, other approaches exist that estimate $Z_s$ based only on porosity [11], grain size [28], bulk density [29], and several others. Although different equations to calculate $Z_s$ produce close results, deviations can range up to ~20% from [9,11] approaches (for sites not shown here). Therefore, error bars of $\pm 25\%$ are added to the $Z_s$ results to account for the different sources of uncertainties mentioned, while the deviations between $Z_s$ and $Z_i$ observed without accounting for the uncertainties were only 12–15%, 4%, and 7% for the three sites, respectively (Figures 6–8). While 25% is acknowledged as a significant uncertainty, the actual results suggest quite acceptable uncertainties in a range quite common and accepted in geotechnical seabed characterization especially following the 20% deviations observed from the choice of method alone, and up to 14% from lab testing alone [27]. The fitting was assumed adequate since we achieved values well within the uncertainty of impedance derived from the sediment samples. Without further direct measurements of impedance (which were not available to us in this study), there is no way to assess the fitting performance beyond this. Since the assessed sediment depths are shallow and sediment properties were rather consistent over these depths, the derived impedance value should be considered approximately constant over the investigated depth, i.e., being representative of the consistent sediment properties. Therefore, the results should be viewed as representative of the investigated depths. This will likely change for future deployments with more layering or deeper investigations. The test sites were located in shallow water environments ranging between 2 and 4.3 m. Ref. [30]

investigated such depths previously applying the multiple reflection approach on acoustic data suggested by [31], where the rends agreed, but the reflection coefficients calculated did not seem to match the reflection coefficients estimated based on sediment properties. It was hypothesized that the deviation resulted from that the approach by [31] was developed for deeper water depths. The inversion approach used in this study performed well overall in the shallow water environments, producing acceptable results despite the uncertainties and limitations discussed earlier. Based on the estimated $Z_i$, porosity and bulk density can then be deduced, in addition to other properties such as water content, void ratio using phase diagram relationships and ultimately, consolidation properties and relative density. All these properties are difficult to estimate in upper sediment layers where dynamic changes are likely to occur on a more frequent basis and high-quality sediment samples are difficult to collect.

Acoustic data is commonly correlated with sediment properties such as porosity and void ratio [9,11,24,32]. Similarly, relationships with strength parameters such as undrained shear strength have been proposed, but rarely reported [30,33]. A direct relationship to $s_u$ and friction angles or relative density is attractive for many engineering and naval applications, as well as the study of local geomorphodynamics. PFFP data represents a convenient pathway to increase joint geotechnical and geoacoustic data sets for the development of possible correlations. Figure 8a suggests a relationship between the PFFP deceleration values with the acoustic impedance, whether it was deduced from sediment properties or from CHIRP sonar, where PFFP deceleration and $Z$ increased with grain size. Despite the limited data points, a linear trend appears feasible between deceleration and acoustic impedance based on the sediment type. Both the CHIRP and PFFP measurements are dependent on geotechnical sediment properties such as bulk density and porosity and strength-related properties, therefore a correlation between both measurements is theoretically expected. $Z$ was insensitive to changes in $s_u$ ranging between 0.4–1.7 kPa. This appears reasonable since this range is limited and includes very soft sediments only. This work limits the analysis to the upper 30 cm, as a result, it does not test the ability of the recommended inversion approach framework to capture differences in acoustic impedance within deeper sediment layers. Future work should target layered stratigraphy, where sediment properties vary along sediment depth to validate the ability of this approach to reflect different sediment properties in layered sediment stratigraphy as well as a correlation to the relative density for coarse-grained sediments, for a wider range of undrained shear strength or plasticity parameters for fine-grained sediments.

Inversion is a common technique for the processing of seismic data and has also gained attention regarding the processing of CHIRP sonar data [32]. Other successful approaches determine the reflection coefficient and apply a seafloor classification model that relates the reflection coefficient to sediment properties, e.g., [9,31]. Ref. [31] presented a pathway to calculate the reflection coefficient from relations of two way travel times and the amplitudes of the CHIRP signal at two reflectors. Ref. [5] explored this method for a CHIRP sonar data set collected in the Potomac River for similar shallow-water conditions and shallow seabed investigation. While general trends between CHIRP sonar results and geotechnical testing and sediment characterization matched, the study raised questions regarding the applicability for shallow sediment depths and how to address fine layering. This initial study tests inversion procedures as suggested by [10,14]. Similarly, as in those studies, this study found the inversion procedure to be successful in relating the CHIRP sonar data to the sediment properties derived from core samples, and it expands on those studies by finding the procedure promising even for shallow water and shallow sediment characterization targets and for correlation with geotechnical data. Most recently, machine learning has been proposed to correlate geotechnical in situ data collection and CHIRP sonar data. Ref. [11] used an acoustic inversion process and correlated the results with Cone Penetration Testing through a machine learning algorithm. With more data sets available, it can be envisioned that such a strategy may also succeed in optimizing joint CHIRP sonar and PFFP surveying.

## 5. Conclusions

This paper presents a shallow seabed investigation combining geotechnical and geophysical tools, specifically, a portable free fall penetrometer, geotechnical lab testing of physical samples, and CHIRP sonar measurements, with the long-term goal to better understand and advance predictions of upper geotechnical sediment properties in shallow water environments from acoustic surveying. An inversion approach was applied on the CHIRP sonar data to deduce the acoustic impedance in the upper 30 cm of the seabed, where box core samples were extracted to compare the CHIRP results with the predicted acoustic impedance based on measured sediment properties, being mainly porosity and bulk density. The acoustic impedance profiles matched reasonably; deviations were limited to <15% supporting the ability of acoustic data to capture variations in sediment properties. Portable free fall penetrometer measurements identified very soft fine-grained sediments at S1, mixed but mostly fine-grained very soft sediments at S2, and loose sand at S3. The CHIRP and PFFP data seemed correlated through the acoustic impedance and PFFP deceleration values, based on variations with sediment type. Acoustic impedance did not change with undrained shear strength for $s_u < 2$ kPa. This work offers an initial attempt to quantitatively relate low-frequency geoacoustic data from CHIRP sonar and geotechnical sediment characterization using portable free fall penetrometer testing in shallow water depths and with focus on shallow seabed surface sediments. The proposed inversion approach proved promising for mapping of the variability in seabed sediment conditions that relate to geotechnical properties and represents an initial step to the development of correlations. Quantitative correlations between geoacoustic surveying and geotechnical properties may optimize seabed characterization with regard to available data volumes, costs, efficiency, and environmental impacts. An improved seabed characterization will improve better decision-making and offshore engineering design. The presented approach extends existing work by focusing specifically on shallow seabed sediments and complements the limited number of studies relating geotechnical and geoacoustic data quantitatively, adding to a better understanding of those relationships. The next steps should focus on estimating sediment properties from impedance profiles and validating it with soil testing for a wider range of seabed types. The data and code used to produce this work is publicly available online [16].

**Author Contributions:** Conceptualization, N.S. and R.J.; methodology, N.S. and R.J.; software, R.J. and R.S.; validation, R.J.; formal analysis, N.S. and R.J.; investigation, N.S., R.J. and G.M.; data curation, R.J. and J.E.M.; writing—original draft preparation, R.J.; writing—review and editing, N.S. and J.E.M.; visualization, R.S.; supervision, N.S.; funding acquisition, R.J. All authors have read and agreed to the published version of the manuscript.

**Funding:** This research was funded by the National Science Foundation for funding this work through grant CMMI-1751463 and the Naval Research Lab through grant N00173-19-1-G018 and the Strategic Environmental Research and Development Program of the DOD through SERDP MR21-1265.

**Data Availability Statement:** This study is available on request from the corresponding author. The data presented in this study are available in [9].

**Acknowledgments:** The authors would like to thank Cristin Wright from Virginia Institute of Marine Science and Liz Smith from Virginia Tech for their on-site assistance during field surveys and data collection. The authors acknowledge two anonymous reviewers who provided constructive comments that assisted with the improvement of the article.

**Conflicts of Interest:** The authors declare no conflict of interest.

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
