# Peer review of "Relating Geotechnical Sediment Properties and Low Frequency CHIRP Sonar Measurements"

_remotesensing, doi:10.3390/rs16020241_

Round 1

Reviewer 1 Report (Previous Reviewer 1)

Comments and Suggestions for Authors

I think this paper can be accepted as the authors well address my comments.

Author Response

The authors thank the reviewer for their previous comments and are glad that the reviewer agrees with the revision. The manuscript has clearly improved from the reviewer’s comments

Reviewer 2 Report (Previous Reviewer 3)

Comments and Suggestions for Authors

I thank the authors for revising their manuscript. Please find below some minor corrections that need to be addressed prior to publication.

Line 25, please change “20 %” to “20%” (it wasn’t addressed).

Line 45-46: Please rephrase the sentence and use better English “...sediment depths of about one meter or more of sediment depth.”.

Please use MDPI citation style throughout the text (see Instruction to Authors for the Remote Sensing journal) (it wasn’t completely addressed). Please look at a previously published paper on the Remote Sensing website to understand how to cite papers in the text). From the Instructions to Authors: “...In the text, reference numbers should be placed in square brackets [ ], and placed before the punctuation; for example [1], [1–3] or [1,3]. For embedded citations in the text with pagination, use both parentheses and brackets to indicate the reference number and page numbers; for example [5] (p. 10). or [6] (pp. 101–105).”.

Line 75: Please also write CPT in full.

Line 84: Missing closing parenthesis and full stop

Line 97, please move “Compressed High Intensity Radar Pulse” to line 47 when the term CHIRP first appears.

Author Response

I thank the authors for revising their manuscript. Please find below some minor corrections that need to be addressed prior to publication

  • Thank you for the feedback. The manuscript has clearly improved from the reviewer’s comments. We have addressed the most recent comments as listed below in a newly revised version of the manuscript.

Line 25, please change “20 %” to “20%” (it wasn’t addressed).

  • The text is updated.

Line 45-46: Please rephrase the sentence and use better English “...sediment depths of about one meter or more of sediment depth.”

  • Thank you for the feedback, the sentence is paraphrased to “This study is focused on investigating seabed sediment properties to at a depth of one meter or more below the seabed surface.”

Please use MDPI citation style throughout the text (see Instruction to Authors for the Remote Sensing journal) (it wasn’t completely addressed). Please look at a previously published paper on the Remote Sensing website to understand how to cite papers in the text). From the Instructions to Authors: “...In the text, reference numbers should be placed in square brackets [ ], and placed before the punctuation; for example [1], [1–3] or [1,3]. For embedded citations in the text with pagination, use both parentheses and brackets to indicate the reference number and page numbers; for example [5] (p. 10). or [6] (pp. 101–105).”

  • Thank you for the feedback, the references style was updated and the intext citation as well.

Line 75: Please also write CPT in full.

  • The text is updated.

Line 84: Missing closing parenthesis and full stop

  • The text is updated.

Line 97, please move “Compressed High Intensity Radar Pulse” to line 47 when the term CHIRP first appears.

  • The definition is moved to line 47.

This manuscript is a resubmission of an earlier submission. The following is a list of the peer review reports and author responses from that submission.

Round 1

Reviewer 1 Report

Comments and Suggestions for Authors

In this study, colocated geotechnical and geoacoustic measurements of different seabed sediment types in shallow 20 water environments (< 5 m of water depth) are analyzed. In general, the topic in this paper is very interesting. However, some important contents are not presented. The authors should comprehensively revise their paper before acceptance.

1. The introduction in this paper is too simple. The authors should comprehensively conduct the literature review. The novelty and contribution of this paper should be highlighted.

2. In practice, the multibeam sonar[1], parametric profiler[2], synthetic aperture sonar [3][4] are used to detect the target in sediment. The authors must comprehensively review these techniques.

[1]Hyeonwoo Cho. AUV-Based Underwater 3-D Point Cloud Generation Using Acoustic Lens-Based Multibeam Sonar. IEEE Journal of Oceanic Engineering, 2018, Doi: 10.1109/JOE.2017.2751139

[2] V.A. Voroiun .Investigations of hydroacoustic parametric sub-bottom profiler characteristics, DOI: 10.1109/OCEANS.1999.800225

[3] Zhang. An omega-k algorithm for multireceiver synthetic aperture sonar. Electronics Letters. 2023, Doi: 10.1049/ell2.12859

[4]Yang. An imaging algorithm for high-resolution imaging sonar system. Multimedia Tools and Applications. 2023, Doi: 10.1007/s11042-023-16757-0

3. The sound velocity in sediment is not the same with that in the water. The authors do not consider this in the paper. The reviewer wanders to know how to compensate it.

4. The authors should perform the comparison between their method and traditional methods. With this operation, the readers can easily understand the advantages of authors’ method.

5. In Fig. 2, the flowchart summarizing the steps of inversion approach is presented. However, some descriptions are not presented. Besides, the reviewer wanders to know the objective function in detail. Since the iteration is used, the robustness should be discussed.

Comments on the Quality of English Language

Fine

Reviewer 2 Report

Comments and Suggestions for Authors

Review of "Relating Geotechnical Sediment Properties and Low Frequency Chirp Sonar Measurements"

Chirp (i.e., Low frequency acoustic methods) represents an important method to reveal subaqueous shallow stratigraphy. The present study investigates co--located geotechnical and geoacoustic measurements of different seabed sediment types in a very shallow subaqueous environments (< 5 m of water depth), which is very useful for confirming the ability of acoustic methods to capture the topmost seabed layers. The manuscript has considerable novelty and should be published. I did not detect any problem that can be fixed, so I suggest publication.

Reviewer 3 Report

Comments and Suggestions for Authors

Please find below my remarks and comments regarding manuscript #remotesensing-2677676.

- Affiliations: Please merge the same affiliations and the “Current affiliation” should be an independent one. Please also avoid underlining them.

- As chirp is an acronym, it should be written with capital letters.

- Line 28, please change “<3” to “< 3”.

- Line 28, please change “20 %” to “20%”.

- Please use MDPI citation style throughout the text (see Instruction to Authors for the Remote Sensing journal).

- Please remove any double spaces “  “ throughout the text.

- Line 40, please change to “be able to”.

- Line 57, “...often requires…” please correct the connection with the previous sentence.

- Line 59, suggestion to change “top 1 m” with “up to 1 m”.

- Figure 1, the overview map shown on the top right does not help understand where the sites are located. There are no toponyms.  The Figure should include a frame with coordinates.

- Line 89, CHIRP should be also written for the first time in full, “Compressed High Intensity Radar Pulse”.

- Line 91 and 111, please check the sweep frequencies if they are correct.

- Line 109, convolution requires two signals, one is the source waveform, the other? It is obvious from Figure 2 but the text should be concise.

- The position of equation 1 is not proper and does not follow the text. Also after the equation the same paragraph should continue (no spacing).

- Figure 2 has thick red arrows in some places and in some not. Is there a special meaning for this?

- Lines 117-119, what are the initial values for acoustic impedance? Where there two different layers (media) considered or more? How was this derived? It is not clear from the text whether the values were obtained from the box core samples, but then what would be the purpose of the whole procedure (in this case this should be clearly described). Please elaborate more on this in section 2.1.

- Section 2.1 was this procedure applied with a ready made software or your own? If with a ready made, please provide proper reference.

- Lines 128-130: Please check sentence syntax.

- Line 136 and 144, the methodology section should not refer to the results section (e.g., “as shown in Table 1”). This do not help the interested reader. Please rephrase the text.

- Equation 3, what does the dot between the number and porosity represent? A multiplication? Please correct this or define the symbol in the text.

- Line 165, please add su symbol in parenthesis, so it is defined for the next line.

- Line 190 impedance is 2.76 and in Table 1, 2.78.

- Please avoid using the asterisk as symbol for multiplication throughout the text. There is a middle dot that can be used instead.

- Table 1, an asterisk as a superscript should be placed at 0.51 (for S3) so to refer to the explanation below the Table.

- Line 189, please rephrase the sentence. I understand that impedance was calculated using two different ways and you compared the results and the difference was small. But restated this more clearly and avoid using the word correlation as no correlation was computed or examined.

- Section 3.3 / 4, please comment on the fitting procedure. Are there any criteria or statistics that were set for ending the iterative procedure? Is the fitting considered adequate? For example, in Figure 6a, a difference of up to 100% can be seen in some values. The fitting seems worse in the beginning and better after 1 ms. Does this affect the results?

- Figure 6b, any comment on the black line deviation?

- Figure 8, please check symbols: blue circle/triangle without fill, black circle/triangle without fill.

- References:

Albatal and Stark 2018 was not found in the references.

Stark et al. 2014b was not found in the references – this also implies that there is a 2014a that does not exist.

ASTM D6913/D6913M (2017) is not referenced in the text.

Jaber, R, & Stark N. Forthcoming. - no year is given in the references list.

Jaber et al. 2023 and Jaber et al. 2022 are given in the text but cannot confirm them in the references list.

Maurya and Sarkar 2021 does not exist in the references list.

Wang et al. 2015 has only two authors and should be changed to Wang and Stewart 2015 in the text.

White, D. J., O’Loughlin, C. D., Stark, N., & Chow, S. H. (2018) is not cited in the text

Please do a full check of the manuscript for missing references.

Comments on the Quality of English Language

Please see my comments and suggestions.
